# Label-Free Quantitative Proteomics Reveal the Involvement of PRT6 in *Arabidopsis thaliana* Seed Responsiveness to Ethylene

**DOI:** 10.3390/ijms23169352

**Published:** 2022-08-19

**Authors:** Xu Wang, Marlène Davanture, Michel Zivy, Christophe Bailly, Eiji Nambara, Françoise Corbineau

**Affiliations:** 1Biologie des Semences, UMR 7622, IBPS, Sorbonne Université, 4 Place Jussieu, 75005 Paris, France; 2GQE-Le Moulon, PAPPSO, INRAE, CNRS, AgroParisTech, Université Paris-Saclay, 91190 Gif-sur-Yvette, France; 3Department of Cell & Systems Biology, University of Toronto, 25 Willcocks Street, Toronto, ON M5S 3B2, Canada

**Keywords:** *Arabidopsis thaliana*, seed dormancy, N-end rule pathway, ethylene, quantitative proteomics, label-free

## Abstract

In *Arabidopsis thaliana*, the breaking of seed dormancy in wild type (Col-0) by ethylene at 100 μL L^−1^ required at least 30 h application. A mutant of the proteolytic N-degron pathway, lacking the E3 ligase *PROTEOLYSIS 6* (*PRT6*), was investigated for its role in ethylene-triggered changes in proteomes during seed germination. Label-free quantitative proteomics was carried out on dormant wild type Col-0 and *prt6* seeds treated with (+) or without (−) ethylene. After 16 h, 1737 proteins were identified, but none was significantly different in protein levels in response to ethylene. After longer ethylene treatment (30 h), 2552 proteins were identified, and 619 Differentially Expressed Proteins (DEPs) had significant differences in protein abundances between ethylene treatments and genotypes. In Col, 587 DEPs were enriched for those involved in signal perception and transduction, reserve mobilization and new material generation, which potentially contributed to seed germination. DEPs up-regulated by ethylene in Col included S-adenosylmethionine synthase 1, methionine adenosyltransferase 3 and ACC oxidase involved in ethylene synthesis and of Pyrabactin Resistance1 acting as an ABA receptor, while DEPs down-regulated by ethylene in Col included aldehyde oxidase 4 involved in ABA synthesis. In contrast, in *prt6* seeds, ethylene did not result in strong proteomic changes with only 30 DEPs. Taken together, the present work demonstrates that the proteolytic N-degron pathway is essential for ethylene-mediated reprogramming of seed proteomes during germination.

## 1. Introduction

At harvest, seeds of numerous species are considered dormant when they cannot germinate even under apparently favorable germination conditions [1,2,3,4]. In *Arabidopsis thaliana*, dormant Col-0 seeds have marked difficulty germinating in darkness at temperatures higher than 15 °C [5,6]; however, they are able to germinate at 25 °C under continuous light [7,8]. This dormancy is progressively broken during dry storage and the germination of dormant seeds is promoted after the application of dormancy-breaking agents, such as chilling, gibberellins (GAs), jasmonates, brassinosteroids, reactive oxygen species (ROS) and reactive nitrogen species (RNS) [5,9,10,11]. The hormonal balance between abscisic acid (ABA) and GAs plays an essential role in the regulation of seed dormancy [11,12,13,14,15]. In addition, it has been demonstrated that the plant hormone, ethylene, could also release seed dormancy of numerous species through crosstalk with ABA and GAs and the ABA/GAs balance [16,17,18]. Interestingly, the actions of these dormancy agents are associated with ethylene production [19], as well as by the activation of their responsive genes, such as ethylene responsive factors (ERFs) or even the downstream genes through their own signaling pathway [16,18,20]. In previous works, Wang et al. (2018, 2021) [6,21] demonstrate that ethylene at 50–100 µL L^−1^ promotes the germination of dormant seeds and that this promoting effect was associated with a down-regulation of genes involved in GAs and ABA signaling such as *RGA* (repressor of *ga1-3*), *RGL2* (RGA-like2), and *ABI5* (ABA INSENSITIVE 5), and with a strong decrease in ABA/GA_4_ ratio.

The N-end rule pathway corresponds to the ubiquitin-proteasome system (UPS) mediated protein degradation by targeting a specific N-terminal residue of a protein [22,23]. This proteolytic pathway has three branches, the Arg/N-end rule pathway, the Ac/N-end rule pathway, and the Pro/N-end rule pathway [24,25,26,27,28]. The Arg/N-end rule pathway identifies specific N-degrons through the activity of E3 ligase, namely, PROTEOLYSIS 1 (PRT1) and PRT6, which recognize the aromatic residues (Phe, Trp, Tyr) and the basic residues (Arg, His, Lys), respectively. The N-degrons as listed above can be generated by the proteolytic cleavage such as N-terminal Met excision by methionine aminopeptidase or endoproteolytic cleavage by endopeptidase. It can also be produced by post-translational modification after the deamination of Asn or Gln, and the oxidation of Cys to secondary residues Asp, Glu, Cys-sulfinic acid, respectively. Subsequently, they are arginylated by arginyl-tRNA transferase to generate a peptide initiating with Arg [29,30]. Using an Arg/N-end rule mutant, especially *prt6*, this pathway has been identified to be involved in many aspects of plant development. Firstly, *prt6* exhibits wavy and lobed leaves with deeper serrations and a loss of apical dominancy [31]. Then, one *prt6* mutant allele, *greening after extended darkness 1* (*ged1*) is more tolerant to starvation conditions, such as prolonged darkness, than the wild type [32,33]. More importantly, particular attention was paid to the lower sensitivity of *prt6* seedlings to hypoxia [34,35]. Furthermore, the *prt6* mutant exhibits an increase in susceptibility to pathogens and down-regulation of JA-responsive genes [36]. Curiously, it was proposed that *prt6* mutant in Arabidopsis and reduced expression of *HvPRT6* in barley showed enhanced resistance to various biotic and abiotic stresses [37,38]. In addition, *prt6* was insensitive to an NO donor SNAP in the stomatal closure and seed germination [39]. Until now, the reports concerning *prt6* in seed germination have described enhanced sensitivity to ABA and NO, impaired seed reserve mobilization and higher tolerance to hypoxia [34,39,40].

In order to improve our knowledge of the mechanism of the N-degron pathway impact in plant development, the availability of genome information and the development of technology have facilitated the omic studies to investigate the protein turnover and detect the half-life of the N-termini [32,34,41,42,43]. Microarray analysis of Choy et al. (2008) [32] shows the decrease in transcripts encoding seed storage proteins (SSPs), oleosins, and late embryogenesis abundant (LEA) proteins in seedlings of *ged1* compared with the wild type. Gibbs et al. (2011) [34] found that the up-regulated genes in *prt6* mutant and *ate1 ate2* mutant share more than half of the 49 core genes from hypoxia response [44]. Zhang et al. (2015) [42] undertook a quantitative proteomic analysis of *prt6* in Arabidopsis roots using terminal amine isotope labeling (TAILS), but this work indicated that the mutation of *prt6* does not markedly enrich the accumulation of destabilizing residues. Recently, N-termiomics has been used again by Zhang et al. (2018) [43] to analyze etiolated seedlings of *prt6*, which identified few up-regulated SSPs and several down-regulated proteases, some of which are regulated by ERFVII [43]. To date, Arabidopsis seeds have not yet been selected to launch omic analysis of the N-end rule pathway, though Gibbs et al. (2011) [34] included seed materials of *prt6* along with the microarray analysis of seedlings. In the present study, we aim to determine the impact of PRT6 E3 ligase and ethylene treatment on the Arabidopsis seed proteome. Using the *prt6* mutant, we previously demonstrated that ethylene could not break seed dormancy of *prt6* [6,21]; therefore, a proteomic method was used to compare the different sensitivity of *prt6* and wild type (Col) seeds to ethylene and we analyzed the proteomic changes in both genotypes incubated at 25 °C in air and in the presence of ethylene.

## 2. Results

### 2.1. Effects of Ethylene Treatment on the Germination of Dormant Seeds

At harvest, seeds from both genotypes (Col and *prt6*) were dormant and did not germinate at 25 °C and in darkness [6]. Seed incubation in the presence of ethylene at 100 µL L^−1^ promoted the germination of Col seeds, while the *prt6* seeds were insensitive to ethylene (Figure 1). Continuously in the presence of this hormone, dormant Col seeds started to germinate after 30 h, and about 55% and 96% of the seed population harvested in 2016 became able to germinate after 48 and 72 h, respectively. Seeds harvested in 2016 germinated better at 25 °C without ethylene (11.9%) than those harvested in 2018 (0%), they could be considered less dormant (Table 1). In addition, Table 1 shows that breaking of dormancy required an application of ethylene for at least 24–30 h. When the batch was deeply dormant like those harvested in 2018, only 7.7% of the seed population can germinate after 16 h of incubation in the presence of ethylene, while a pretreatment for 30 h resulted in 54.6% germination. Except for 48 h-application of ethylene, seeds harvested in 2016 which were less dormant than those harvested in 2018, were more sensitive to ethylene; half of the population was able to germinate in air at 25 °C after a 24 h or 30 h pretreatment with ethylene, respectively, for the batch harvested in 2016 and 2018 (Table 1).

### 2.2. Ethylene-Induced Proteome Changes Observed at 30 h but Not 16 h after Treatments

To identify the global proteomic changes in *prt6* and Col seeds in response to ethylene, proteome data were collected from dry seeds and imbibed seeds for 16 h and 30 h in the presence or absence of ethylene. A total of 1737 proteins were detected across samples from dry and 16 h-imbibed seeds and 1002 of them were selected for further quantitative analysis by integration of the extracted ion current (XIC) (Appendix A). Two-way ANOVA analysis showed that no differentially expressed proteins (DEPs) were present in these proteins (adjusted *p* value < 0.05) (Figure 2A). In 30 h-imbibed seeds, 2552 proteins were identified and 2202 proteins were selected for further analysis (Appendix A). Except for 38 out of 1002 quantified proteins that were only present in the 16 h-proteome, all the remaining identified proteins were also present in the 30 h-proteome (Appendix A). After 30 h, 619 proteins showed a significant interaction between the genotype and the ethylene treatment, indicating that the two genotypes responded differently to ethylene (adjusted *p* value < 0.05) (Figure 2B) (Appendix A).

Scatter plots (Figure 2C,D) illustrated that ethylene did not significantly affect the majority of proteins of both genotypes in 16 h-imbibed seeds (Figure 2C). However, in the 30 h-imbibed seeds, it is obvious that ethylene affected the abundance of a large number of proteins in Col seeds, while ethylene had only a limited effect on the seed proteome of the *prt6* mutant (Figure 2D). Protein expression levels for each replicate were assessed using PCA analysis (Figure 2E,F). In the “dry” and “16 h” proteomes, we noticed that *prt6* was more predominantly located at the positive side of PC1 (13% of the variation) while Col was at the negative side of PC1. Furthermore, PC2 was able to separate dry seeds from 16 h-imbibed seeds, but the four conditions of seed imbibition could not be separated as shown by the clustered ellipses (Figure 2E). However, after 30 h of imbibition, each condition was clearly separated in the plot (Figure 2F). It was also shown that the PC1 from 30 h-imbibed seed proteome was 46.8% that was higher than 13.3% from “dry” and “16 h” proteomes.

### 2.3. Effects of Ethylene on the Expression of DEPs

As shown in Figure 2B, 619 proteins showed genotype/ethylene interaction. Quantification of these DEPs on the basis of log XIC values in the four genotype/treatment combinations (Col+, Col−, *prt6*+, *prt6*−) was used to present expression patterns by a hierarchical clustering heatmap (Figure 3A). It was revealed that Col+ clustered separately from others. The sample clustering fits nicely with the observation that Col seeds, but not others, could germinate in the presence of ethylene. On the other hand, the color of each column indicated that ethylene had a more pronounced effect in Col than in *prt6*. Tukey analysis (*p* value < 0.05) and fold change filtering (fold change >1.3 or <0.7) were then employed to select DEPs that showed a significant effect of ethylene in Col and in *prt6* (Appendix A). In Col+/Col−, 311 down-regulated and 276 up-regulated DEPs were identified, while we identified only 15 down-regulated and 15 up-regulated DEPs in *prt6+/prt6**−* (Figure 3B,D). In addition, 23 down-regulated and 77 up-regulated DEPs show genotype-dependent differences in protein abundances in the absence of ethylene (Figure 3C) (Appendix A).

A significant number of DEPs from the comparison groups, Col+/Col− and *prt6*+/*prt6*−, were found in the SeedNet, a transcriptional topological model [45] (Figure 4A,B). It was revealed that 234 out of 311 down-regulated DEPs and 234 out of 276 up-regulated DEPs in Col+/ Col− were included in the SeedNet model (Appendix A). As shown in Figure 4A, down-regulated DEPs were predominantly located within region 1 where genes were associated with dormancy. Conversely, up-regulated DEPs were primarily presented within region 3 where genes were associated with germination. These results indicated that the regulation of the protein abundances by ethylene in wild-type (Col) seeds were consistent with the general transcriptional regulation related to seed germination. In parallel, 10 out of 15 down-regulated DEPs and 13 out of 15 up-regulated DEPs in *prt6+/prt6−* were also found in the SeedNet (Figure 4B) (Appendix A). As shown in Figure 4B, although the numbers were small, up-regulated DEPs were located within regions 2 and 3, while down-regulated DEPs were in regions 1 and 3. The Venn diagram showed 29 DEPs belonging to both groups Col+/Col− and *prt6*+/*prt6*−, which only PHYTOCHROME A (AT1G09570) decreased in *prt6* after ethylene treatment (Figure 4C). The relative abundance of the 29 common DEPs were used to present expression patterns by a hierarchical clustering heatmap (Figure 4D). Four clusters were divided according to the expression profiles. Cluster I illustrated the reduction in protein expression in Col and *prt6* in the presence of ethylene. In contrast, Cluster IV showed that ethylene enhanced the protein expressions in both genotypes, even the reduction and induction were much more pronounced in Col than that in *prt6*. Cluster II included two proteins SAD6 (stearoyl-acyl carrier protein 9-desaturase, AT1G43800) and an adenine nucleotide alpha hydrolases-like superfamily protein (AT3G03270), both of which showed more protein in *prt6*− than in Col− in the absence of ethylene and both were up-regulated by ethylene. Cluster III presented the opposite changes in which ethylene increases the expressions of the five proteins LDAP2 (lipid droplet associated protein 2, AT2G47780), EXP2 (expansin 2, AT5G05290), GLL22 (GDSL lipase-like protein 22, AT1G54000), JLL30 (jacalin-related lectin, AT3G16420) and PYK10 (beta-glucosidase, AT3G09260) in Col, while the five proteins were down-regulated in *prt6* with ethylene treatment. We noticed that the five proteins were also increased in *prt6* in the absence of ethylene in comparison with Col.

GO enrichment analysis was performed to interpret the functional modules of DEPs. Down- and up-regulated DEPs in Col+/ Col− group were analyzed separately. As shown in Figure 5 [46,47], the network of biological process enrichment analysis indicated that down-regulated DEPs were enriched in the seed development and dormancy, secondary metabolic process, glutathione metabolism, various responsive processes, such as the response to ABA, lipid, water, light, heat and H_2_O_2_. While up-regulated DEPs were enriched in the amino acids biosynthetic process, gene expression and protein metabolism, RNA modification, lipid catabolic process and also some responsive processes, such as response to ABA, alcohol and H2O2. To sum up, the signal perception and transduction from down and up-regulated proteins, the energy mobilization from up-regulated proteins, new material generation from up-regulated proteins and some other assisting processes work together to contribute to seed germination of Col seeds.

### 2.4. Effect of prt6 on the Expression of DEPs in the Absence of Ethylene after 30 h of Imbibition

As shown in Appendix A, 100 DEPs were selected in the group *prt6*−/Col−. This group was selected in order to determine the impact of the mutation of *prt6* in seed dormancy and proteome, independently of the regulation by ethylene. Table 2 shows, except for 19 proteins which were miscellaneous, classified DEPs from this group. Three proteins encoded by “core hypoxia” genes characterized by transcriptionally up-regulation in hypoxia conditions [44], increased in *prt6* mutant in the absence of ethylene for 30 h (*prt6*−). Among them, the protein level of HB1 (hemoglobin 1, AT2G16060) and SAD6 (stearoyl-acyl-carrier-protein desaturase family, AT1G43800) were around 20 times higher in *prt6*− than in Col−. In addition, we noticed that several proteins involved in nucleobase synthesis, transcription, translation, protein modification and transport were moderately increased in *prt6*−, most of which showed an increase of less than two times. Three seed storage proteins LEAs were found with less abundance in *prt6*− compared to Col−. Other proteins involved in ABA synthesis, such as AO4 (aldehyde oxidase 4, AT1G04580) was in reduced abundance in *prt6,* when proteins involved in methionine metabolism, such as MTO3 (S-adenosylmethionine synthetase, AT3G17390) and MAT3 (methionine adenosyltransferase 3, AT2G36880) had higher abundance in *prt6*.

### 2.5. Functional Class Scoring Analysis without Filtering DEPs in 30 h-Proteome

Because of the low variance in the comparison group, especially in *prt6*+/*prt6−*, fewer DEPs were achieved after cut-off selection, so over-representation analysis failed to give us an informative result. GSEA (gene set enrichment analysis) [48] and Pathifier [49] were then used to overcome this limitation. The subtle but coordinated changes in the group *prt6*+/*prt6−* were captured by GSEA analysis (Figure 6). Most of the up-regulated BP (Biological Process) terms were enriched in protein synthesis, transport, localization and modification. The most significantly enriched BP term represented by the highest count was “peptide biosynthetic process”. Some BP terms related to tissue development or morphogenesis (class 2, Figure 6) were improved in the presence of ethylene, but at the same time, BP terms involved in development growth (class 5, Figure 6) were inhibited in the presence of ethylene. These results suggested that BP terms in class 5 might partially lead to the dormancy of *prt6* when treated with ethylene. In parallel, GSEA analysis was also carried out in Col+/Col− group to compare the different effects of ethylene between *prt6* and Col (Appendix A). It seems that lipid transport and localization, and response to hydrogen peroxide were down-regulated in both groups. However, some BP terms, such as fatty acid oxidation, lipid biosynthetic process, mitochondrial transport, hormone biosynthetic process coenzyme biosynthetic process, were specific in the Col+/Col−, which contributed to the diversity and complexity of the affected BP categories. It was shown that even without ethylene treatment, the genotype effect had revealed some distinct BP terms (Appendix A). We can also notice that the term, lipid localization, was down-regulated in both Col+/Col− and prt6−/Col− groups (Appendix A).

As shown in Figure 6 and Appendix A, the enrichment of lipid localization in both genotypes could not distinguish which one (Col or *prt6*) was more affected by ethylene. Therefore, Pathifier analysis was chosen to give us a quantified deregulation level of a specific biological process pathway. At the same time, some categories that are known to be associated with seed germination or some related to the function of PRT6 were also included, such as hormone regulation, protein metabolism (folding, localization), molecule transport, development growth and response to hypoxia were selected to perform Pathifier analysis with Col− acting as a reference.

As shown in Figure 7A, Pathifier successfully discriminated between dormant and non-dormant seeds. Among the four samples, only Col+ possessed germination potential, and PDS values of each BP category were particularly higher than that in the other three samples. Secondly, some biological processes were deregulated between Col− and *prt6*−, especially the BP term, response to hypoxia, which was much more affected than others. In addition, with the treatment of ethylene, response to H_2_O_2_, response to stress were also deregulated in *prt6* to some degree, but the inductive effect could not be compared with the changes observed in Col in the presence of ethylene (Figure 7B).

### 2.6. Effect of prt6 on the Fate of Storage Proteins during Germination

Abundant proteins in seeds such as seed storage proteins (SSPs), late embryogenesis abundant proteins (LEAs) and oleosins were selected to compare their changes in abundance as related to ethylene treatment or the genotype. In 30 h-proteome, 47 abundant proteins were identified while in the dry and 16 h-proteome, the quantity was only 36, all of which were present after 30 h-proteome (Appendix A). The common 11 SSPs in dry seeds, 16 h- and 30 h-imbibed seeds, and their abundance ratios (*prt6*−/Col−) (Table 3) showed that cruciferin A (AT5G44120.3) and 2S albumins (AT4G30880) were most significantly up-regulated in the *prt6* in dry seeds when compared with Col dry seeds, suggesting that PRT6 was involved in the stabilization of some SSPs during seed development. After 16 h of imbibition, levels of all SSPs were slightly higher in *prt6*, but after 30 h, most were not altered in their protein abundance, indicating that during seed imbibition, the N-end rule affected the mobilization of SSPs.

In 30 h-proteome, the effect of ethylene was also compared between wild type and *prt6* mutant. It was shown that overall ratios of Col+/Col− were much lower than that of *prt6*+/*prt6*− (Appendix A), indicating that the *prt6* mutation probably inhibited ethylene-mediated degradation of SSPs, LEA and oleosins.

### 2.7. Hormone-Related Proteins in Seed

Proteins participating in hormone synthesis or hormone signaling were obtained from proteomic data. Four proteins involved in the ethylene synthesis were identified: MTO3 (S-Adenosylmethionine Synthase 2, AT3G17390), MAT3 (Methionine Adenosyltransferase 3,AT2G36880) and SAM (S-Adenosylmethionine Synthetase 1, AT1G02500) which catalyzed the formation of S-adenosylmethionine, and ACO (ACC oxidase 1, AT2G19590) which oxidized ACC in ethylene biosynthesis.

As shown in Figure 8, ethylene significantly resulted in an increase in protein level of MTO3 and MAT3 in Col. The abundances of ACO and SAM1 also increased to some degree in Col (although not significantly according to ANOVA analysis). However, ethylene did not markedly affect the four proteins in *prt6.* AO4 (Aldehyde Oxidase 4) catalyzed the final steps of carotenoid catabolism, while PYR (Pyrabactin Resistance) acted as one of the ABA receptors [50,51]. In Col, ethylene dramatically increased the abundance of PYR and decreased the abundance of AO4. The decrease in AO4 was consistent with the decrease in endogenous ABA in Col seeds when treated with ethylene [21]. On the other hand, the increased abundance of PYR was not expected, since it is involved in seed dormancy [52]. No significant difference in protein abundance was observed between *prt6−* and *prt6+* for AO4 and a slight increase was observed for PYR.

## 3. Discussion

In the present work, a proteome analysis was performed to investigate the role of PRT6 in the ethylene response during seed germination. We demonstrated that only the application of ethylene longer than 30 h was able to break the dormancy of Arabidopsis seeds placed at 25 °C (Table 1). Longer imbibition durations (30 h) also allowed us to observe the interaction effects between ethylene treatment and genotype (Figure 2). We aimed to find some genes that could rescue the insensitivity of *prt6* seeds to ethylene, so the five genes in cluster III (Figure 4D) were selected. However, germination tests showed that the five mutants germinated as well as Col in the presence of ethylene (data not shown), suggesting that one single gene was not sufficient to affect seed responsiveness to ethylene, or there might be due to functional redundancy.

The proteome data were analyzed using two classes of bioinformatics tools. Firstly, GO over-representation analysis was carried out using the defining DEPs. However, over-representation analysis could not be powerful when we identified only a small number of DEPs, like in our comparison group *prt6*+/*prt6−*. The second class of tools such as GSEA acted as a complementary method that could detect small but consistent changes in a set and then yielded the affected categories by using two enrichment analyses [53,54]. Furthermore, Pathifier belonging to the second class was also employed to improve the depth and breadth of our study.

Compared to leaf proteome or entire plant proteome, seed proteome is characteristic to over-represent the general categories such as SSPs, cell rescue, cell defense and protein fate [55]. For this reason, several selected GO terms such as response to stress, protein folding, protein transport and protein metabolism were included in the Pathifier analysis (Figure 7). At the same time, SSP analysis was launched by filtering all these proteins to compare their exact values (Table 3 and Appendix A). One BP term, response to hypoxia, was also subjected to the Pathifier analysis, given that the effect of the N-end rule coordinated with oxygen sensing in the regulation of seedling survival or root growth under hypoxia [34,35,56,57,58,59,60]. This would fit nicely with our result that the PDS score of response to hypoxia is significantly different between *prt6*− and Col−, meaning that many proteins involved in this process were affected in *prt6* mutant seeds (Figure 7B). Pathifier analysis evaluated the deregulation level of a specific pathway in the tested condition compared with the control, but it failed to indicate the direction of this change in comparison with control, that is, up-regulation or down-regulation. However, GSEA analysis was specialized to access the differential expression of gene sets for core genes in significant functional categories from a ranked list, which could meet the requirement to a certain degree (Figure 6 and Appendix A). For example, the GSEA visualization plot of the BP category, lipid localization in the group *prt6*−/Col− showed that the leading edge gene set presented at the bottom of the ranked list. This suggested that this category was mostly suppressed in the *prt6*− mutant in the absence of ethylene (Appendix A). As shown in Appendix A, all the proteins involved in lipid localization in 30 h-proteome were collected, which consisted of different classes of proteins, such as oleosins, lipid transfer proteins or some other binding proteins. It was shown that the reduced protein levels predominated through the list (Appendix A). Considering that more than half of the proteins here were storage proteins or oleosins, it seemed logical that they are degraded and serve as energy support. In this case, many more carrier proteins or transfer proteins would be required to transport the “cargo”, but in Appendix A, there are few aiding proteins (AT3G05420, AT5G42890) with a ratio higher than 1. The predominance of the down-regulated proteins in this category might result from the lower abundance of transfer proteins. In the over-representation analysis of DEPs in Col+/Col−, we also found some BP terms concerning lipid metabolism, but which were more related to lipid catabolic process and fatty acid beta-oxidation (Figure 4). It is also true for oleosins and oil body-associated proteins, most of which ratios in Col+/Col− were generally lower than that in *prt6*+/*prt6−* (Appendix A). This indicated that in the presence of ethylene, the PRT6 mutation impaired the oil body metabolism, as the same phenotype was observed in the mutant seedlings [40,61].

SSPs accumulated during seed development could supply energy for the early stage of seed germination. SSPs are the most abundant proteins in the total seed proteome and the presence of extremely abundant SSPs decreases the sensitivity to detect other low abundant proteins in seeds. In our study, we demonstrated that the N-end rule pathway could regulate the accumulation or stabilization of SSPs during seed development or seed imbibition (Table 3). It is possible that the insensitivity of *prt6* seeds to ethylene is related to the inhibited SSPs mobilization after 30 h (Appendix A). It was reported that the abundance of SSPs was increased, especially cruciferins in *prt6* etiolated seedlings (4 d under green light) [43]. Whereas, Choy et al. (2008) [32] reported that transcript abundance of genes encoding SSPs, oleosins, and LEA proteins, were much lower in the *ged1* (an allele of *prt6*) seedlings (5 d in the dark followed by 2 d in the light). The opposite changes of SSPs at the transcript level to the protein level might be due to the different conditions of light in the two studies, as it was reported that the N-end rule coordinated with light perception to regulate seedling establishment after germination [62].

It was well documented that seed germination and dormancy were regulated via crosstalk of ABA, Gas and ethylene, whose hormone synthesis and signaling pathways could interact with each other to control seed dormancy [2,3,17,18,63]. The breaking of seed dormancy by exogenous ethylene was associated with activation of its own responsive genes as well as fine-tuning the hormonal balance between ABA and GA [6,16,18,20,21]. Here, the Pathifier analysis was carried out to calculate the deregulation score of seed sensitivity to hormones (Figure 7A and Appendix A). The PDS scores of Col+ were significantly higher than that of Col− (Appendix A). In contrast, with the treatment of ethylene, the PDS scores of response to the three hormones did not show a dramatic increase (Appendix A). Particularly, almost no difference in PDS score of response to ethylene between *prt6−* and *prt6*+, indicating that the related proteins in this category were reluctant to alter in the presence of ethylene. Proteins related to ethylene synthesis such as MTO3, MAT3, ACO and SAM, were enhanced in Col in the presence of exogenous ethylene, which was in agreement with Liu et al. [64] who demonstrate that exogenous ethylene induces endogenous ethylene biosynthesis (Figure 8). Furthermore, there is a negative interaction between ethylene and ABA in seed germination [17,18,21]. In Col, exogenous ethylene decreased ABA level [21] and the protein abundance of AO4, which was involved in ABA synthesis (Figure 8). In addition, the protein levels of PYR were stimulated by ethylene in Col, but considering that ethylene could suppress ABA signaling [65], we speculated that the higher abundance of PYR might be inactivated [66], or a feedback response due to the lower production of ABA in the presence of ethylene.

The N-end rule pathway target proteins to be degraded through proteasome at the N-terminal residues [22]. Considering the diversity of mechanisms of neo-N-termini generation, the targets of the N-end rule pathway were only achieved in the pre-pro-protein, while in plants, they are ERFVII proteins, all of which were starting with Met-Cys [34]. In our study, 100 proteins were significantly accumulated in *prt6* mutant at 30 h (Table 2, Appendix A). All amino acid sequences of the 100 proteins were retrieved from the database to identify new putative substrates among Nt Met-Cys proteins. It showed that only one protein ASN3 (Asparagine Synthetase 3, AT5G10240) was a potential target, which was also suggested by Majovsky et al. (2014) [41] with the increased abundance in both *ate1 ate2* seedlings and *prt6* seedlings. To date, several transcriptomic or proteomic analyses of the effect of the N-end rule pathway by using the *prt6* and *ate1 ate2* mutants have been published [32,34,41,42,43]. As shown in the Appendix A, there were few proteins overlapped, suggesting the difference in protein abundance between seedlings and seeds. Though Gibbs et al. (2011) [34] included imbibed *prt6* seeds, the different imbibition duration and different statistical methods might also contribute to identifying fewer DEPs. It is worth noting that AT2G16060 (Hemoglobin 1) is represented across all the published data.

Here, the *prt6* mutation was connected with ethylene signaling at the level of proteomes, which was a novel aspect further studying the effect of the N-end rule pathway in seed dormancy. In a future study, more attention could be paid to the transcriptomic changes of *prt6* imbibed seeds and the involvement of ERFVIIs in the insensitivity of *prt6* to ethylene.

## 4. Materials and Methods

### 4.1. Seed Materials

*Arabidopsis thaliana* seeds from Columbia-0 (Col-0) were used as the wild type of this study. Mutant *prt6* (SAIL-1278-H11) was in the genetic background of Col-0 and obtained as a gift from Dr. M. J. Holdsworth (Nottingham University). For seed propagation, seeds were incubated in water at 4 °C for 3 days then transferred into the potting mixture (Jiffy^®^ substrates, Tref, Paris, France) and placed in a growth chamber at 21 °C under a photoperiod of 16 h light/8 h dark. Siliques were harvested at maturity in 2016 and 2018, and the seeds collected were stored at −20 °C until experiments.

### 4.2. Germination Assay

Germination assays were performed in darkness in 9-cm Petri dishes with 100 to 200 seeds per assay in 3 replicates. Seeds were placed on a filter paper laid on top of cotton wool moistened with deionized water at 25 °C. Germination assays in the presence of gaseous ethylene were carried out in tightly closed 360 mL containers, in which was injected gaseous ethylene (5%) (Air Liquide, Paris, France) to set the concentration of ethylene at 100 µL L^−1^. A seed was considered to be germinated as soon as the radicle protruded through the seed coat. Germination counts were made every 24 h for 7 days, and the results presented are the means of the germination percentages obtained with 3 replicates ± SD.

### 4.3. Seed Treatment

Col and *prt6* seeds were imbibed on a filter paper placed on the top of cotton wool moistened with deionized water in tightly closed 360 mL containers (50 mg per assay in 3 replicates) in darkness. Seeds were incubated with (+) or without ethylene (−) 100 µL L^−1^ for 16 h and 30 h. They were then frozen in liquid nitrogen and stored at −80 °C. Dry seeds from both genotypes were also analyzed.

### 4.4. Protein Extraction and LC-MS Analysis

The total protein was extracted by a modified phenol extraction method as described by Desjardin et al. (2012) [67]. The protein concentration was determined using 2D-QUANT kit (Amersham Biosciences, Little Chalfont, UK), then it was digested by trypsin (Promega, Madison, WI, United States). Desalting and concentration of the peptides were implemented before the operation of HPLC by NanoLC-Ultra (Eksigent, Dublin, OH, USA) coupled with an MS analysis by Qexactive PLUS (Thermo Fisher, Waltham, MA, USA). LC-MS analyses were performed on 3 independent batches: dry seeds, 16h-imbibed seeds and 30 h-imbibed seeds.

### 4.5. Protein Identification and Quantification

Arabidopsis FASTA format database was downloaded from TAIR (https://www.Arabidopsis.org/, accessed on 1 January 2019) and database searches were performed using the X!Tandem search engine [68] and protein inference and post-treatment filtering was performed by using X!TandemPipeline 02.28 [69]. Data filtering was achieved according to a peptide E-value < 0.01, protein log (E-value) < −4 and a minimum of two identified peptides per protein. Peptides were quantified based on eXtracted Ion Current (XIC) of their peptides using MassChroQ 2.2.12 Black Caïman [70]. Relative quantification of the proteins was calculated using R scripts by analyzing peptide intensities obtained from XIC. Firstly, reliable peptides were selected according to the criteria described in Belouah et al. (2018) [71], i.e., shared peptides, uncorrelated peptides, peptides showing retention time instability, and peptides showing more than 3 missing data out of 12 were filtered out. Only proteins represented by at least two reliable peptides were used for analysis. Protein relative abundances were computed as the sum of the peptide intensities. Protein abundance changes were detected separately for 16 h- and 30 h-imbibed seeds by a 2-way ANOVA (ethylene treatment and genotype) with interaction. The effect of dry seeds was estimated in comparison to 16 h-imbibed seeds in the absence of ethylene treatment. Proteins showing a *p* value corrected for multiple comparisons < 0.05 were considered as showing a significant variation. Among them, the down and up-regulated proteins were then defined as those that significantly changed after Tukey test with *p* value < 0.05 and fold-change < 0.7 or > 1.3.

### 4.6. Bioinformatic Analysis

Principal component analysis was conducted by the R package factoextra [72]. Heatmap clustering was drawn by R package pheatmap [73] with values being centered in the row direction and clustering distance method being chosen as “Euclidean”. Volcano plot was finished by EnhancedVolcano [74] with the cutoff of log 2 (fold change) at 0.4. Down-regulated and up-regulated proteins in the comparison group were input into the SeedNet [45], in the platform of Cytoscape to capture the overlapping proteins. Gene ontology (GO) analysis was launched by the R package clusterProfiler [75]. Biological Process (BP) terms of Arabidopsis were downloaded from the new release (TAIR Data 20180331) of GO annotation (https://www.Arabidopsis.org/index. Jsp, accessed on 1 January 2019) and automatically generated annotations, that is, Inferred from Electronic Annotation (IEA) were removed to improve the credibility of overrepresentation analysis. The redundant BP terms were discarded and by using the function “simplify” in R package clusterProfiler [75] to keep the term with a lower P value among all the similar terms. The connected BP terms were then input into the networkD3 [47] to present the functional module. Gene Set Enrichment Analysis (GSEA) of the comparison group was carried out by analyzing the rank-ordered protein lists based on abundance ratios Col+/Col−, *prt6*+/*prt6*− and *prt6*−*/*Col− on all quantified proteins, significant BP terms were kept after *p* value (< 0.05) selection and discarding the redundant BP terms [75]. Pathifier pipeline was applied to determine the extent to which individual BP terms are deregulated in every individual condition [49]. Before Pathifier analysis, the selected BP terms extracted from GO annotation data were mapped to ancestor BP terms by using the function “buildGOmap” to obtain all the genes included in a specific BP category [75]. Each BP category included a list of proteins whose expression data were available in all samples thus allowing the calculation of a deregulation score (PDS) measuring the deviation of a biological process from a reference condition (Col−).

## Figures and Tables

**Figure 1 ijms-23-09352-f001:**
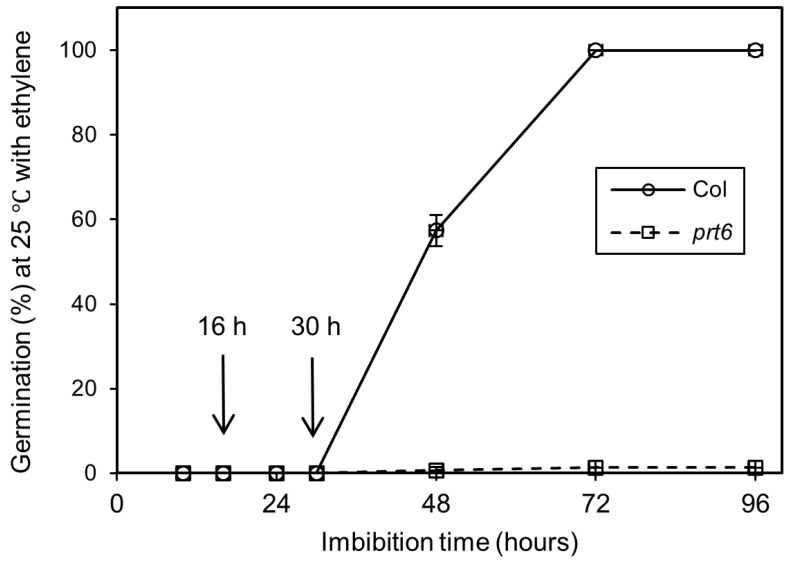
Germination percentages obtained with Col and *prt6* seeds placed at 25 °C on water in the presence of ethylene at 100 µL L^−1^. Means of 3 replicates ± SD. Seeds harvested in 2016. The arrows indicate the sampling time used for label-free quantitative analysis.

**Figure 2 ijms-23-09352-f002:**
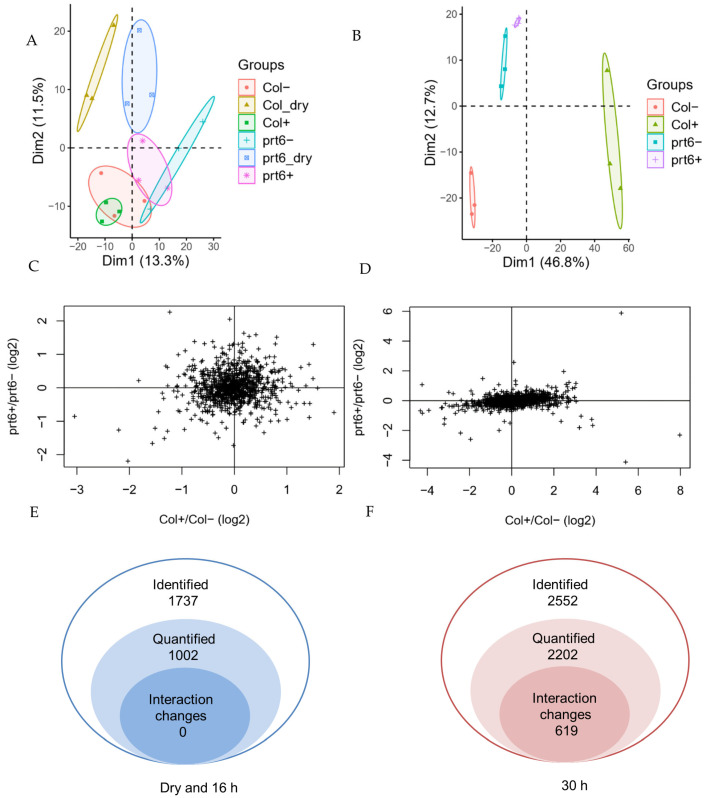
Effects of ethylene on the 16 and 30 h-proteome in Arabidopsis seeds (Col and *prt6*). (**A**,**B**): The number of identified, quantified and significantly changed proteins in label-free quantification by the XIC method from seed proteome of dry seeds and 16 h-imbibed seeds (**A**) and 30 h-imbibed seeds (**B**). (**C**,**D**): Scatter plots of the log 2 transformed ratio (Col+/Col− versus *prt6+/prt6−*) for 1002 proteins from “dry and 16 h” proteomes (**C**) and 2202 proteins from 30 h-proteome (**D**), + represents the presence of ethylene treatment, − Represents the absence of ethylene treatment. (**E**,**F**): Principal components analysis (PCA) of the protein quantification for 1002 proteins from proteomes in dry and 16 h-imbibed seeds (**E**) and 2202 proteins from 30 h-imbibed seed proteome (**F**), each dot corresponds to one of the replicates, each circle indicates 95% confidence interval of each condition. Seeds harvested in 2016.

**Figure 3 ijms-23-09352-f003:**
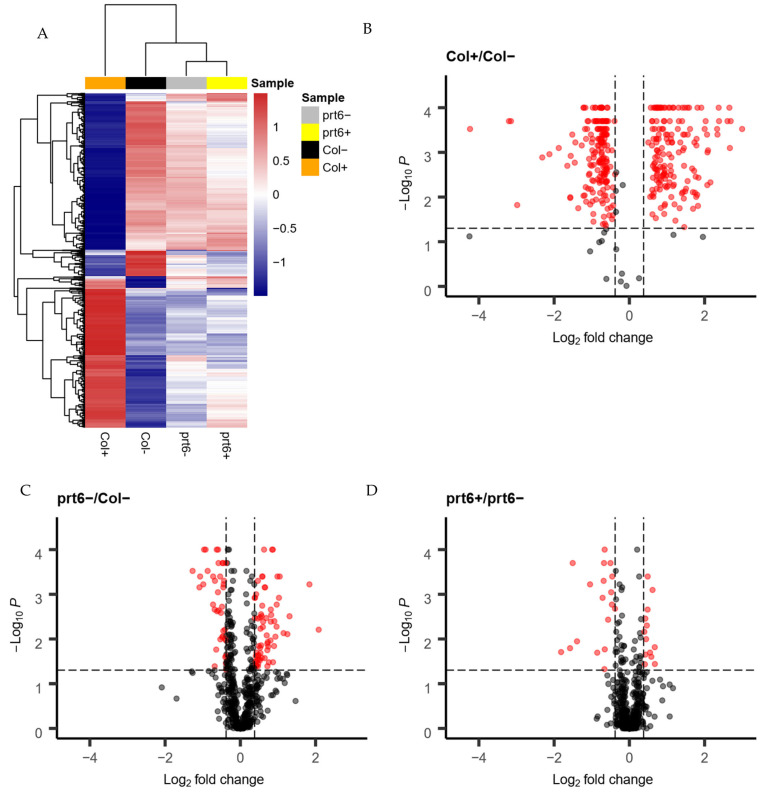
Expression patterns of the 619 DEPs estimated by XIC abundance. (**A**): The hierarchical clustering analysis of the 619 DEPs in Col+, Col−, *prt6*+, *prt6*−. In the heatmap, blue represents down-regulated expression, red represents up-regulated expression and white represents no change in expression. (**B**–**D**): The volcano plots representing the quantitative analysis of 619 proteins in Col+ versus Col− (**B**), *prt6*− versus Col− (**C**), *prt6*+ versus *prt6*− (**D**). *X*-axis and *Y*-axis represents the log 2 (fold ratio) and −log 10 (*p*), respectively. Red dots indicate that the proteins have significant differences (*p* < 0.05, and fold change < 0.7 or fold change > 1.3), but the threshold of fold change was shown as |log2 (fold ratio)| > 0.4 in the plot), while the black dots correspond to proteins with no significant differences. The horizontal dashed line represents *p* = 0.05, namely, −log 10 (*p*) = 1.3. The two vertical dashed lines correspond to log 2 (fold change) = 0.4. Seeds harvested in 2016.

**Figure 4 ijms-23-09352-f004:**
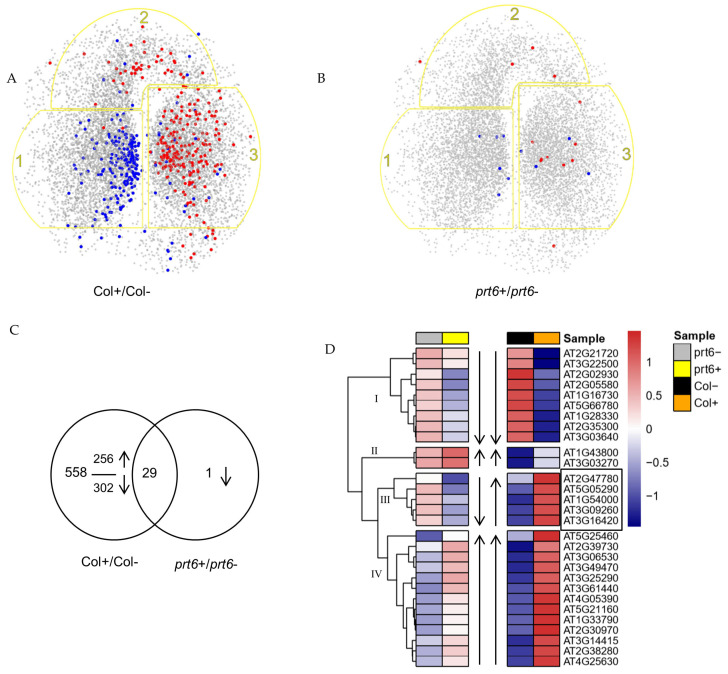
Classification of DEGs from comparison groups, Col+/Col− and *prt6*+/*prt6*−. (**A**,**B**): Localization of the DEPs from group Col+/Col− (**A**) and *prt6+/prt6− (***B**) in the SeedNet gene co-expression network. Yellow outlines indicate as follow: region 1, genes associated with seed dormancy; region 3, genes associated with seed germination; and region 2, genes associated with dormancy-germination transition. Red dots and blue dots represent down and up-regulated proteins in corresponding group. (**C**): A Venn diagram showing the common and specific proteins in groups, Col+/Col− and *prt6*+/*prt6−*, up and down-arrows indicate up and down-regulated proteins, respectively. (**D**): A hierarchical clustering of the 29 common DEPs in Col+, Col−, *prt6*+, *prt6−*. The top bars show the clustering of the different samples, *prt6−* (grey), *prt6+* (yellow), Col− (black) and Col+ (orange). In the heatmap, blue represents down-regulated expression, red represents up-regulated expression and white represents no change in expression. The inner arrows represent the effect of ethylene on *prt6* and Col with showing up-arrow if fold change ratio (Col+/Col− or prt6+/prt6*−*) < 1, down-arrow if fold change ratio (Col+/Col− or prt6+/prt6−) > 1. Gene trees were divided into 4 clusters indicated by I, II, III, IV. Seeds harvested in 2016.

**Figure 5 ijms-23-09352-f005:**
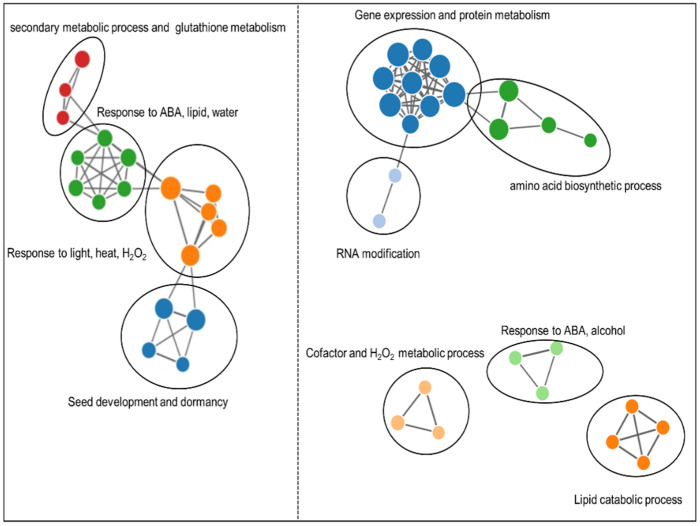
Enrichment map organizing enriched terms of biological process from DEPs in Col+/Col−, into a network with edges connecting overlapping gene sets. The left part represents enrichment map from down-regulated proteins in Col+/Col−, while the right part represents enrichment map from up-regulated proteins in Col+/Col−. Each node corresponds to significant biological process, and the size of the node indicates gene numbers in each set, while the edge connects the overlapping gene set. Distinct colors are given to different clusters calculated by walktrap community finding algorithm [46,47].

**Figure 6 ijms-23-09352-f006:**
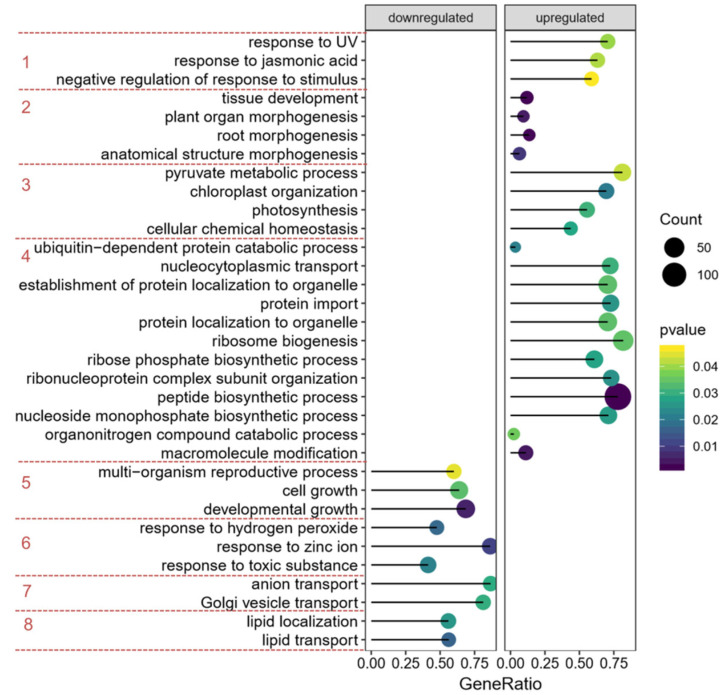
A dot plot of enriched BP terms in *prt6*+/*prt6−* determined from GSEA results. The size of the dot represents count of the core proteins in each set, the color represents *p* value, and GeneRatio represents the number of core genes in gene set/the total number of genes in this gene set. The enriched BP terms were classified into several groups depending on functional connection and similarity, that is, class 1, response to stimuli; class 2, plant morphogenesis; class 3, energy metabolism; class 4, protein synthesis, localization and modification; class 5, development growth; class 6, response to stimuli; class 7, anion and Golgi transport; class 8, lipid transport and localization.

**Figure 7 ijms-23-09352-f007:**
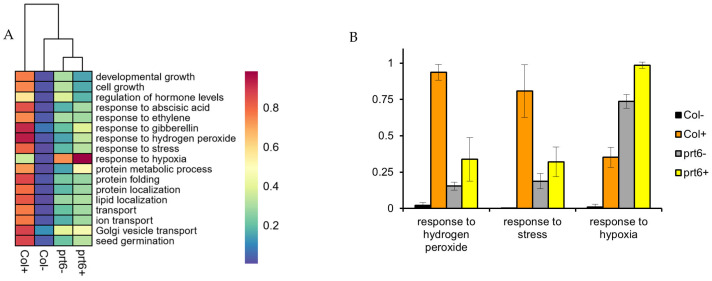
Individual pathway deregulation scores (PDS) of the selected biological process in 4 samples showing by heatmap clustering (**A**) and bar plot (**B**). In the heatmap, blue to red represents the PDS score from low to high.

**Figure 8 ijms-23-09352-f008:**
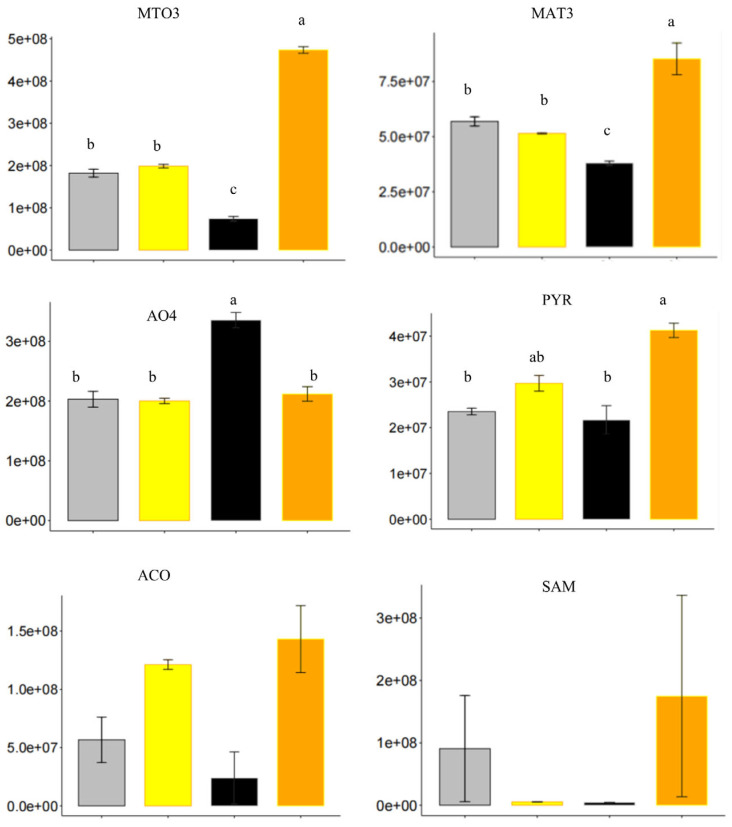
Quantification of the proteins involved in the hormone synthesis and signaling in 4 samples, Col− (black), Col+ (orange), *prt6*− (grey), *prt6*+ (yellow). Significant levels were only shown on the proteins which passed the 2 factors ANOVA analysis and have interaction effects.

**Table 1 ijms-23-09352-t001:** Effects of the duration of incubation at 25 °C in the presence of ethylene at 100 µL L^−1^ on subsequent germination of dormant Col seeds after transfer in air at 25 °C. Means of 3 replicates ± SD Seeds harvested in 2016 and 2018.

Duration of Seed Incubation in the Presence of Ethylene (h)	Germination (%) after 7 Days at 25 °C in Air after Ethylene Treatment	
	2016 Harvest	2018 Harvest
0	11.9 ± 4.0	0
10	39.9 ± 3.0	-
16	48.3 ± 4.0	7.7 ± 7.5
24	54.5 ± 6.2	21.0 ± 1.0
30	-	54.6 ± 4.6
48	61.9 ± 3.7	81.3 ± 4.0

**Table 2 ijms-23-09352-t002:** Classified DEPs from comparison group *prt6*−/Col−. Except for 19 proteins which were miscellaneous, 80 out of 100 significant proteins in group *prt6*−/Col− are shown below.

ID.	*Description*	*prt6−/Col−*
	**Hypoxia-responsive**	
AT1G19530	DNA polymerase epsilon catalytic subunit A	1.64
AT2G16060	Hemoglobin 1	18.74
AT1G43800	Plant stearoyl-acyl-carrier-protein desaturase family protein	21.64
	**Carbohydrate metabolism**	
AT4G33820	Glycosyl hydrolase superfamily protein	0.39
AT5G63800	Glycosyl hydrolase family 35 protein	0.48
AT3G09260	Glycosyl hydrolase superfamily protein	5.78
AT1G53580	Glyoxalase II 3	1.52
AT4G34260	1,2-alpha-L-fucosidase	0.70
AT5G26120	Alpha-L-arabinofuranosidase 2	0.65
AT4G15210	Beta-amylase 5	1.98
AT4G36360	Beta-galactosidase 3	0.60
AT5G49360	Beta-xylosidase 1	1.51
AT1G02640	Beta-xylosidase 2	1.47
AT5G57550	Xyloglucan endotransglucosylase/hydrolase 25	0.47
AT2G47510	Fumarase 1	1.30
AT1G12780	UDP-D-glucose/UDP-D-galactose 4-epimerase 1	1.57
AT2G39730	Rubisco activase	1.80
	**Cell wall**	
AT1G19900	Glyoxal oxidase-related protein (involved in cell wall modification)	0.51
AT5G15490	UDP-glucose 6-dehydrogenase protein (required for the formation of cell wall ingrowths)	1.47
AT5G05290	Expansin A2	4.24
AT3G14310	Pectin methylesterase 3	1.32
	**Oxidation and reduction**	
AT3G61070	Peroxin 11E (integral to peroxisome membrane, controls peroxisome proliferation)	1.34
AT2G46750	D-arabinono-1,4-lactone oxidase activity	0.51
AT3G14415	Aldolase-type TIM barrel family protein (modulates ROS-mediated signal transduction)	1.95
AT4G35000	Ascorbate peroxidase 3	1.67
AT4G20860	FAD-binding Berberine family protein (involved in the generation of H_2_O_2_)	0.53
AT2G02930	Glutathione S-transferase F3	0.52
AT2G30870	Glutathione S-transferase PHI 10	1.95
AT2G47730	Glutathione S-transferase phi 8	2.14
AT5G66920	SKU5 similar 17 (oxidation-reduction process)	1.63
AT3G27820	Monodehydroascorbate reductase 4	1.83
AT1G60680	NAD(P)-linked oxidoreductase superfamily protein	0.66
AT3G44880	Pheophorbide a oxygenase family protein with Rieske 2Fe-2S domain-containing protein	1.38
AT5G63030	Thioredoxin superfamily protein	0.67
	**Transcription**	
AT5G02490	Heat shock protein 70 (HSP 70) (mediator of RNA polymerase II transcription subunit 37c)	1.38
AT4G25630	Ibrillarin 2 (Mediator of RNA polymerase II transcription subunit 36a)	1.45
AT3G08030	DNA-directed RNA polymerase subunit beta	1.77
AT3G03270	Adenine nucleotide alpha hydrolases-like superfamily protein	7.91
AT1G74260	Purine biosynthesis 4	1.61
AT3G57610	Adenylosuccinate synthase (purine synthesis)	1.50
AT2G45300	RNA 3’-terminal phosphate cyclase/enolpyruvate transferase%2C alpha/beta	1.80
AT3G52150	RNA-binding (RRM/RBD/RNP motifs) family protein	1.37
AT3G19130	RNA-binding protein 47B	1.86
AT5G63420	RNA-metabolising metallo-beta-lactamase family protein	1.34
AT3G58510	DEA(D/H)-box RNA helicase family protein	1.31
	**Translation**	
AT4G01310	Ribosomal L5P family protein	1.35
AT5G23900	Ribosomal protein L13e family protein	1.38
AT3G63490	Ribosomal protein L1p/L10e family	1.44
AT1G50920	Nucleolar GTP-binding protein ( Involved in the biogenesis of the 60S ribosomal subunit)	1.57
ATCG00830	Ribosomal protein L2	1.67
AT2G33800	Ribosomal protein S5 family protein	1.67
AT1G32990	Plastid ribosomal protein l11	1.91
AT2G40290	Eukaryotic translation initiation factor 2 subunit 1	1.39
AT3G56150	Eukaryotic translation initiation factor 3C	1.74
AT4G20360	RAB GTPase homolog E1B (Elongation factor Tu)	1.55
	**Protein folding**	
AT3G13470	TCP-1/cpn60 chaperonin family protein (protein folding)	1.49
AT1G56340	Calreticulin 1a (Molecular calcium-binding chaperone promoting folding)	2.38
AT1G09210	Calreticulin 1b (Molecular calcium-binding chaperone promoting folding)	2.07
AT3G09440	Heat shock protein 70 (HSP70) (mediate the folding of newly translated peptides)	1.48
AT1G55490	Chaperonin 60 beta (suppressing protein aggregation in vitro)	1.34
AT3G16420	PYK10-binding protein 1 (chaperone that facilitates the correct polymerization of PYK10)	4.71
	**Protein transport**	
AT4G10480	Nascent polypeptide-associated complex (NAC) 2C alpha	1.50
AT3G49470	Nascent polypeptide-associated complex subunit alpha-like protein 2	1.40
AT3G15980	Coatomer subunit beta -3	1.44
AT3G46830	RAB GTPase homolog A2C	1.44
	**Protein modification**	
AT4G17040	CLP protease R subunit 4	1.38
AT2G47390	Prolyl oligopeptidase family protein	1.46
AT2G38280	AMP deaminase%2C putative / myoadenylate deaminase	1.34
AT3G45010	Serine carboxypeptidase-like 48	3.57
AT5G10240	Asparagine synthetase 3	1.45
AT5G11880	Pyridoxal-dependent decarboxylase family protein	1.33
AT1G61790	Oligosaccharyltransferase complex/protein N-linked glycosylation	1.41
AT1G48630	Receptor for activated C kinase 1B (shuttle activated protein kinase C )	2.04
AT3G18130	Receptor for activated C kinase 1C (shuttle activated protein kinase C )	1.66
AT1G25490	ARM repeat superfamily protein (encoding phosphoprotein phosphatase)	1.69
	**LEAs**	
AT3G22500	Seed maturation protein (LEA protein)	0.63
AT3G51810	Stress induced protein (LEA protein)	0.69
AT5G66780	Late embryogenesis abundant protein	0.55
	**Hormone**	
AT1G04580	Aldehyde oxidase 4 (ABA synthesis)	0.61
AT3G17390	S-adenosylmethionine synthetase family protein (ethylene synthesis)	2.45
AT2G36880	Methionine adenosyltransferase 3 (ethylene synthesis)	1.50

**Table 3 ijms-23-09352-t003:** The protein expression ratios (*prt6*−/Col−) of SSPs in dry seeds, and in seeds imbibed for 16 and 30 h.

Locus	Description	0	16	30
AT5G44120.3	Cruciferin A	1.44	2.73	0.56
AT5G44120.1	Cruciferin A	1.13	1.74	0.88
AT1G03880	Cruciferin B	1.12	1.43	0.81
AT4G28520	Cruciferin C	1.00	1.04	0.73
AT4G30880	2S albumin	1.59	1.42	0.90
AT3G22640	Cupin	0.97	1.14	0.73
AT1G07750	RmlC-like cupins	0.73	1.65	1.05
AT4G36700	RmlC-like cupins	0.77	1.19	0.81
AT1G03890	RmlC-like cupins	0.85	1.01	0.75
AT2G28490	RmlC-like cupins	0.94	1.09	0.83
AT2G18540	RmlC-like cupins	1.07	1.17	0.85

## Data Availability

All data in this study can be found in public data bases and Appendix A, as described in the Materials and Methods section.

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
