# Peer review of "Label-Free Quantitative Proteomics Reveal the Involvement of PRT6 in Arabidopsis thaliana Seed Responsiveness to Ethylene"

_ijms, 2022, doi:10.3390/ijms23169352_

Round 1

Reviewer 1 Report

Dear authors, I will proceed to give you my opinion and feedback about your manuscript.

Introduction:

- Check double spaces, typos and °C.

Results:

2.1. Effects of ethylene treatment on the germination of dormant seeds:

- Are the percentages corresponding to the hours after germination? “Germination of dormant Col seeds was initiated after 30 h in the presence of ethylene, and about 55% and 96% of the seed population harvested in 2016 became able to germinate after 48 and 72 h, respectively” If so, please rewrite the sentence to make it clearer.

- Fig 1. Which is the reason why the seeds from 2016 are less dormant than the ones from 2018? This needs an explanation because the germination rate after 48 hours is logical showing less germination in the oldest population. How did the authors determine the dormancy of the seeds?

2.2. Ethylene-induced proteome changes observed at 30 h but not 16 h after treatments

- Which batch of seed had been used, 2016 or 2018? It is not indicated in the text nor in figure 2. In addition, the authors use dry seeds as a kind of control. Are these seeds new or are collected in 2016 or 2018? This information must be clarified to better understand the experimental settings.

-Why do the authors only use dry seeds for 16h and not for 30h?

2.3. Effects of ethylene on the expression of DEPs

- If the mutant prt6 is less sensitive to ethylene treatment what is the purpose of using the mutant for this experiment? Did the authors check seeds from 2016 or 2018 of this mutant in order to see the effect in old seeds?

- I would like to know the expression levels of the genes from the 15 down and up-regulated DEPs from the comparison prt6+/prt6-. This could give the authors more information about what is happening at a molecular level in the mutant treated with ethylene and discuss better the importance of PRT6 E3 ligase in the ethylene metabolism and seed germination. In line with these, anatomical studies of the embryos will bring light to this study.

- “fold change filtering (fold change >1.3 or < 0.7) was then employed to select DEPs that showed a significant effect of ethylene in Col and in prt6” Why there is a change in the restriction factor?

- Why are the authors not showing the comparison between prt6+/Col+ in the VolcanoPlot in Fig3?

- Which are the criteria the authors used for doing the clusters in Fig 4?

2.4 Effect of prt6 on the expression of DEPs in the absence of ethylene after 30 h of imbibition

- “These proteins were classified manually since GO terms were uninformative (Table 2)” The GO is not uninformative, if you do not have enough information then the GO will not be supported. The authors must correct this sentence and explain which are the criteria used for building a heatmap without being supported by the software? Without GO terms, how do the authors know the group of genes/proteins in each group?

- A general comment: the authors must be clear in the writing because sometimes the reader does not know if they are referring to genes or proteins, transcriptomics, and proteomics. They mix the terms in a confusing way.

- Why did the authors decide to use the data from the group prt6−/Col− to do a GO analysis? Is this set of data the most interesting for determining the impact of PRT6 E3 ligase and ethylene treatment on the Arabidopsis seed proteome?

2.5 Functional class scoring analysis without filtering DEPs in 30 h-proteome

- Could the authors add the meaning of BP in the text? Otherwise, the reader has to wait until the materials and methods to know what is it.

- “Curiously” and “Surprisingly” are not appropriate terms. Change them.

- Fig 7 B what is it showing? There is no information about the meaning of the graph. Is it showing transcription information? Response to hydrogen peroxide, to stress and hypoxia what are they, GO, group of genes, treatment done in seeds? This must be clarified.

2.6 Effect of prt6 on the fate of storage proteins during germination

- I suggest to the authors to do staining of oleosins in the seeds/embryos with and without ethylene will help to understand the results obtained in the proteome.

- Measurement of ethylene or an ethylene marker should be done in the mutant if the authors want to say this “ the prt6 mutation inhibited ethylene-mediated degradation of SSPs, LEA and oleosins”.

2.7 Hormone related proteins in seed

- Fig8 I would like to know why the authors are expressing the protein levels in box plots. What do they want to represent them? Are these graph proteins or genes? The samples are from 2016 or 2018 seeds?

-Maybe the ethylene did not have an effect at a protein level but it had at a transcriptomic level. The authors should check the gene expression of MTo3, MAT3, AO4, ACO, SAM and PYR to analyze a possible post-transcription effect which explains better the hormonal status of the seeds and the effects on the proteome.

Conclusions:

-The authors say in the introduction “we aim to determine the impact of PRT6 E3 ligase and ethylene treatment on the Arabidopsis seed proteome.” But in the conclusions, they do not answer the question.

- “In the present work, a proteome analysis was performed to investigate the role of Arg/N-end rule pathway in the ethylene response during seed germination” The authors are making mention of the Arg/N-end rule pathway but they have not brought any information about the lifetime of the proteins, if there is any anomaly, how common is for the proteins they have DEPs to be ubiquitinated. This part of the paper is not clear, and they claim it in the title. I suggest a change of title for another more adequate to the main message of the paper.

- General comment: The authors did a proteome of seeds and they describe the results obtained by using different bioinformatical approaches but after all, what is the biological relevance of this study? This paper is lacking biological explanation.

Author Response

Comments and Suggestions for Authors

Dear authors, I will proceed to give you my opinion and feedback about your manuscript.

Reply to reviewer 1

Dear Reviewer 1,
First thank you a lot for your comments concerning the paper we want to publish in IJMS.
You will find below our reply (in blue) to your questions and we hope that they will convince you.

Introduction:

- Check double spaces, typos and °C. We have checked the spaces, typo and °C Results:
2.1. Effects of ethylene treatment on the germination of dormant seeds:

- Are the percentages corresponding to the hours after germination? “Germination of dormant Col seeds was initiated after 30 h in the presence of ethylene, and about 55% and 96% of the seed population harvested in 2016 became able to germinate after 48 and 72 h, respectively” If so, please rewrite the sentence to make it clearer. Figure 1 shows the germination percentages calculated after seed sowing in the presence of ethylene; seeds were continuously in the presence of the gas. We have modified the sentence in the text.

- Fig 1. Which is the reason why the seeds from 2016 are less dormant than the ones from 2018? This needs an explanation because the germination rate after 48 hours is logical showing less germination in the oldest population. How did the authors determine the dormancy of the seeds?
Table 1 (and not Figure 1) shows the effect of the duration of application of ethylene on the subsequent germination obtained 7 days after seed transfer in air at the same temperature (25°C). Experiment was carried out with seeds freshly harvested (in 2016 and 2018) and stored at -20°C until the experiment in order to maintain the dormancy, then both populations can be considered to have the same age (see section 4.1 of Materials and methods). Seeds are considered as dormant when they cannot germinate in darkness at 25°C (section 4.2 of Materials and methods), while they easily germinated at 10°C (refs 5 and 6).

Without ethylene, seeds harvested in 2016 germinated better at 25°C (11.9%) than those harvested in 2018 (0%); they can then be considered as less dormant (Table 1). In addition, except for 48 h-ethylene application, they are more sensitive to ethylene.
We have modified the text of the section 2.1 to be clearer.

2.2. Ethylene-induced proteome changes observed at 30 h but not 16 h after treatments

- Which batch of seed had been used, 2016 or 2018? It is not indicated in the text nor in figure 2. In addition, the authors use dry seeds as a kind of control. Are these seeds new or are collected in 2016 or 2018? This information must be clarified to better understand the experimental settings.
The seed batch used for studying the proteome changes was harvested in 2016. This information was indicated in the legends of Figures 2 and 3.

-Why do the authors only use dry seeds for 16h and not for 30h?

Since the proteomic study indicated that the proteome is not affected by seed imbibition during 16 h either in the presence or the absence of ethylene, and that a two-way ANOVA analysis shows that no differentially expressed proteins were present in 16 h-imbibed seeds compared to dry seeds, we have limited our study on 30 h-imbibed seeds with prt6 and Col imbibed with and without ethylene (4 groups indicated in Figure 2F).

Materials for seed proteome analysis were prepared two twice. The first time, we prepared dry and 16 h-imbibed seeds, but very few significant proteins were achieved, we speculate that the duration is not long enough to allow

us to see the difference. Besides, we try to focus our study on the effect of ethylene on seed germination during the imbibition process, rather the effect of imbibition on the seed proteome changes.

2.3. Effects of ethylene on the expression of DEPs

- If the mutant prt6 is less sensitive to ethylene treatment what is the purpose of using the mutant for this experiment? Did the authors check seeds from 2016 or 2018 of this mutant in order to see the effect in old seeds? As indicated at the beginning of section 2.3, 619 proteins showed genotype/ethylene interaction (Figure 2B), the aim of this study was to present the quantification and expression of these DEPs by using a hierarchical clustering heatmap (Figure 3A). Although ethylene did not improve the germination of the prt6 mutant, 15 down-regulated and 15 up-regulated DEPs were identified in the presence of ethylene, and 23 down-regulated and 77 up-regulated DEPs were identified in the absence of ethylene.
The analysis of DEPs identified allows us to determine their localization in the SeedNet co-expression NetWork, i.e. in the regions 1, 2 and 3 characterized respectively with expression of genes associated with seed dormancy (1), the dormancy-germination transition (2), and seed germination (3). These results are original and new.

In addition, we checked the seed germination in the presence or absence of ethylene in several batches, not only with seeds harvested in 2016 and 2018, prt6 was always insensitive to ethylene, but for seed proteomic analysis was only done with seeds harvested in 2016.

- I would like to know the expression levels of the genes from the 15 down and up-regulated DEPs from the comparison prt6+/prt6-. This could give the authors more information about what is happening at a molecular level in the mutant treated with ethylene and discuss better the importance of PRT6 E3 ligase in the ethylene metabolism and seed germination. In line with these, anatomical studies of the embryos will bring light to this study.

As listed in the supplementary dataset S2, the annotation of the 30 DEPs showed that most of them were seed storage proteins or housekeeping proteins. We also agree that it will be interesting if we could get the gene expressions of the 30 genes in prt6+/prt6. We should have done that at that time. Although seed dormancy could be maintained in -20°C few years, 6 years at -20°C would decrease seed dormancy. Then, if we use these seeds now, firstly, prt6 might be able to respond to ethylene to extent at 25 °C, and the gene expression profiles of the 15 genes might also be different.

Various treatments, such as gibberellins and stratification, could break seed dormancy of prt6 that showing protruded radicle (Wang et al., 2018), but ethylene could not break seed dormancy without protruded radicle, so the anatomical structure of the ungerminated prt6 embryos might be similar with that of ungerminated Col.

- “fold change filtering (fold change >1.3 or < 0.7) was then employed to select DEPs that showed a significant effect of ethylene in Col and in prt6” Why there is a change in the restriction factor?

In transcriptomic analysis, it is popular to use fold changes >2 or <0.5 as a restriction factor, but in proteomic study, this restriction factor was often compromised,

- Why are the authors not showing the comparison between prt6+/Col+ in the VolcanoPlot in Fig3? Technically, it is easy to show prt6+/Col+ in the VolcanoPlot, but its biological meaning is complicated, since there are two factors involved, genotype and ethylene treatment. Then, to be clearer, we have not done theVolcanPlot for prt6+/Col+.

- Which are the criteria the authors used for doing the clusters in Fig 4

Firstly, 29 proteins were selected from the overlapped proteins in Figure 4C, then the heatmap cluster was done by R script, pheatmap, with log 10 modified XIC data as matrix, scale='row', cluster_cols = FALSE, clustering_distance_cols = "euclidean" and border_color=TRUE.

2.4 Effect of prt6 on the expression of DEPs in the absence of ethylene after 30 h of imbibition

- “These proteins were classified manually since GO terms were uninformative (Table 2)” The GO is not uninformative, if you do not have enough information then the GO will not be supported. The authors must correct this sentence and explain which are the criteria used for building a heatmap without being supported by the software? Without GO terms, how do the authors know the group of genes/proteins in each group?

When we run GO analysis by webpage or R package, we could normally achieved some enriched GO terms with a certain amount of random gene IDs. Normally, the more we input, the more chance we can succeed. In our

case, we only have 100 genes IDs as input, we tried with different kinds of webpage kit or R package, including, “Clusterprofiler” used in this study. The common result is that they yielded several GO terms but less than half of the genes were detected in our samples with GO terms even with good gene ratio, which means that most of genes could not be annotated by GO. Similar results were also found by Zhang et al., (2018).

- A general comment: the authors must be clear in the writing because sometimes the reader does not know if they are referring to genes or proteins, transcriptomics, and proteomics. They mix the terms in a confusing way. We agree and have corrected the text.
- Why did the authors decide to use the data from the group prt6/Colto do a GO analysis? Is this set of data the most interesting for determining the impact of PRT6 E3 ligase and ethylene treatment on the Arabidopsis seed proteome?

We selected the group prt6-/Col- in order to determine the impact of the N-end rule pathway, i.e. of the PRT6 E3 ligase, in dormancy and proteome, independently of the regulation of the germination.by ethylene.

2.5 Functional class scoring analysis without filtering DEPs in 30 h-proteome

- Could the authors add the meaning of BP in the text? Otherwise, the reader has to wait until the materials and methods to know what is it.
We have added the meaning of BP in line 6 of the section 2.5

- “Curiously” and “Surprisingly” are not appropriate terms. Change them.

We have changed or suppress these terms in the text of section 2.5., and in all the manuscript.

- Fig 7 B what is it showing? There is no information about the meaning of the graph. Is it showing transcription information? Response to hydrogen peroxide, to stress and hypoxia what are they, GO, group of genes, treatment done in seeds? This must be clarified.
This graph was achieved by a modified Pathifier analysis (Drier et al., 2013, ref. 49). Pathifier pipeline was applied here to determine the extent to which individual GO-BP terms are deregulated in every individual condition. Before Pathifier analysis, the selected BP terms extracted from GO annotation data were mapped to ancestor BP terms by using the function buildGOmapto get all the genes included in a specific BP category. Each BP category included a list of proteins whose expression data were available in all samples thus allowing the calculation of a deregulation score (PDS) measuring the deviation of a biological process from a reference condition (Col).

2.6 Effect of prt6 on the fate of storage proteins during germination

- I suggest to the authors to do staining of oleosins in the seeds/embryos with and without ethylene will help to understand the results obtained in the proteome.
We agree that it will be interesting to do oleosin staining, however, as we can see from supplementary datasetS3, oleosin proteins encoded by 7 genes were quantified. Most of the ratios fluctuated around 1. We did noticed a few ratios smaller than 0.5, but there were also some ones bigger than 2. As the oleosin staining was not able to differentiate the specific oleosins and it detected the oleosins as a whole in a semi-quantitative manner, so we are wondering whether is will bring us new information.

- Measurement of ethylene or an ethylene marker should be done in the mutant if the authors want to say this “ the prt6 mutation inhibited ethylene-mediated degradation of SSPs, LEA and oleosins”.
As far as we know, endogenous ethylene or ACC measurement was tricky by HPLC, let alone our samples were the tiny Arabidopsis seeds. If we could find which new genes (except for ERFVIIs) was disturbed by PRT6 in the ethylene transduction pathway or the ethylene downstream responsive pathway, that result in the abnormal degradation of SSPs, LEA and oleosins, it will be a novel project. Here we just open the questions and wait for the following study.

We want to say that the prt6 mutation resulted in an inhibition of degradation of SSPs, LEA and oleosins, but that it was not necessary associated with ethylene synthesis.

2.7 Hormone related proteins in seed

- Fig8 I would like to know why the authors are expressing the protein levels in box plots. What do they want to represent them? Are these graph proteins or genes? The samples are from 2016 or 2018 seeds?

The sample is from 2016, and Figure 8 corresponded to the level of proteins.

Boxplot here was used to show the abundance of hormone related proteins. May be this representation was not suitable since it required more than 20 replicates, then as you suggested we have changed the Figure 8 using a line graph.
-Maybe the ethylene did not have an effect at a protein level but it had at a transcriptomic level. The authors should check the gene expression of MTO3, MAT3, AO4, ACO, SAM and PYR to analyze a possible post- transcription effect which explains better the hormonal status of the seeds and the effects on the proteome.

We agree that it will be interesting if we could get the gene expressions of MTO3, MAT3, AO4, ACO, SAM and PYR. We should have done that at that time. However, although seed dormancy could be maintained at -20 °C for few years, 6 years in the fridge would result in a gradual decrease in seed dormancy. If we use these seeds now, firstly, prt6 might be able to respond to ethylene to some extent at 25°C, and the gene expression profiles of the 15 genes might also be different.

Conclusions:

-The authors say in the introduction “we aim to determine the impact of PRT6 E3 ligase and ethylene treatment on the Arabidopsis seed proteome.” But in the conclusions, they do not answer the question.

We agree that our paper mainly presented the changes in proteome profile in seeds of Col and prt6 mutant placed in the absence and presence of ethylene. However, our data demonstrate clearly the role of PRT6, and then of the N-end rule pathway, on the proteome and various biological process.

- “In the present work, a proteome analysis was performed to investigate the role of Arg/N-end rule pathway in the ethylene response during seed germination” The authors are making mention of the Arg/N-end rule pathway but they have not brought any information about the lifetime of the proteins, if there is any anomaly, how common is for the proteins they have DEPs to be ubiquitinated. This part of the paper is not clear, and they claim it in the title. I suggest a change of title for another more adequate to the main message of the paper.

We agree your comments and we have changed the tittle to Label-free Quantitative Proteomics Reveal the Involvement of PRT6 in Arabidopsis thaliana Seed Responsiveness to Ethylene”,

However, it is known that the substrates of PRT6 can be generated by the proteolytic cleavage such as N- terminal Met excision by methionine aminopeptidase or endoproteolytic cleavage by endopeptidase, but also by post-translational modification. In addition, degradation also depends on the structural and subcellular context of the destabilizing residue, so it will be difficult to trace back the potential substrates from the global snapshot of the protein lifetime measurement.

- General comment: The authors did a proteome of seeds and they describe the results obtained by using different bioinformatical approaches but after all, what is the biological relevance of this study? This paper is lacking biological explanation.
We agree with you, that this study was mainly a description of the changes in seed proteome using bioinformatics analysis. However this paper was the first to demonstrate the role of PRT6 in the regulation of seed dormancy and seed proteome by ethylene. In addition, the data presented complete the results previously published by the authors in IJMS in 2018 [6] and in J Integr Plant Biol in 2021 [21].

We thank you again for your review and we hope that our reply to your questions and comments, and the correction of some parts of the text will allow the publication of this work.

Best regards Francoise Corbineau Sorbonne Université

Reviewer 2 Report

I liked your work very much, you can see in it a lot of analytical work that you put in to prove your theses. The genome-wide studies of Arabidopsis thaliana and the way your results are presented are noteworthy. The presented tables and figures are clear, legible and transparent. You can learn a lot by analyzing them. The selection of references is sufficient and proves your knowledge of the issue under study. The results of your research may find practical application in commercial agriculture. I will recommend the Editorial Board to accept your work for publication without any changes.

Author Response

Comments and Suggestions for Authors from Reviewer 2

I liked your work very much, you can see in it a lot of analytical work that you put in to prove your theses. The genome-wide studies of Arabidopsis thaliana and the way your results are presented are noteworthy. The presented tables and figures are clear, legible and transparent. You can learn a lot by analyzing them. The selection of references is sufficient and proves your knowledge of the issue under study. The results of your research may find practical application in commercial agriculture. I will recommend the Editorial Board to accept your work for publication without any changes.

Reply

Dear Colleague,

Thank you so much for your comments concerning the work that we want to publish in IJMS. We will now correct the manuscript taking into account the remarks of Reviewer 1.

Best regards Françoise Corbineau Emeritus Professor
